# Micro Transit Simulation of On-Demand Shuttles Based on Transit Data for First- and Last-Mile Connection

Cristian Poliziani [1,*], Gary Hsueh [2], David Czerwinski [3], Tom Wenzel [1], Zachary Needell [1], Haitam Laarabi [1], Joerg Schweizer [4] and Federico Rupi [4]

1   Lawrence Berkeley National Laboratory, 1 Cyclotron Rd, Berkeley, CA 94720, USA
2   CHS Consulting Group, 1617 Clay Street, Oakland, CA 94612, USA
3   Department of Marketing and Business Analytics, San José State University, One Washington Square, San José, CA 95192, USA
4   Department of Civil, Chemical, Environmental, and Materials Engineering, University of Bologna, Viale del Risorgimento 2, 40136 Bologna, Italy
*   Correspondence: cpoliziani@lbl.gov; Tel.: +1-650-539-7730

**Abstract:** We simulate the introduction of shared, automated, and electric vehicles (SAEVs) providing on-demand shuttles service in a large-scale transport digital twin of the San Francisco Bay Area region (California, USA) based on transit supply and demand data, and using the mesoscopic agent-based Behavior, Energy, Autonomy, and Mobility beta software (BEAM) developed at the Lawrence Berkeley National Laboratory (LBNL). The main goal of this study is to test the operations of this novel mobility service integrated with existing fixed-route public transportation service in a mesoscopic simulation of a real case scenario, while testing the BEAM beta software capabilities. In particular, we test the introduction of fleets of on-demand vehicles bound to operate within circular catchment areas centered on high-frequency transit stops, with the purpose of extending the reach of fixed-route transit by providing an alternative first- and last-mile connection at high-frequency public transport stations. Results show that on-demand automated shuttles represent the best solution for some users, increasing the overall transit ridership by 3%, and replacing mostly ride-hail trips, especially those connecting to transit stops, but also some walking trips. This type of service has the potential to reduce overall vehicle miles traveled (VMT), increase transit accessibility, and save energy, but future research is needed to optimize this type of service and make it more attractive to travelers.

**Keywords:** micro transit; accessibility; transport; transit data; simulation

## 1. Introduction

The main weakness of an existing public transport network is the poor proximity of the home locations of potential users to transit stops or light rail and rail stations, particularly in low-density areas, such as Santa Clara County, California, where fixed-route local transit services are inefficient and generate unsustainable ridership, as it is possible to observe from the NTD (National Transit Database: https://www.transit.dot.gov/ntd, accessed on 14 April 2023)— for example, the average agency cost per passenger in 2021 was USD 27.01 for the Santa Clara Valley Transportation Authority (VTA) operating in the Santa Clara County (570 inhabitants per square Km) and USD 10.84 for the San Francisco Municipal Transportation Agency (SFMTA) operating in the City of San Francisco, California, where transit is operating in a denser area (7124 inhabitants per square Km). Hess (2009) observed that the transit accessibility—which quantifies how convenient, effective, and easy is to access a specific transport mode for residents—greatly affected the mode choice, especially for older people [1]. As described by Farber and Grandez (2017) for the city of Toronto, the accessibility to transit considerably affects its efficacy [2]. Zuo et al. (2020a) find that transit service can be greatly extended by improving first- and last-mile access to transit, and disadvantaged residents, who are the most likely to consider public transit, receive better

and more equitable transit accessibility to jobs than others [3]. Kanuri et al. (2019) argue that first- and last-mile connectivity is an important factor in enabling greater integration and accessibility to mass transit networks for the largest number of urban residents [4]. For these reasons, authorities are addressing the problem of the first- and last-mile connection for a public transport network as a method to improve the accessibility of the public transport service and hence its level of usage.

Micro transit, or transit service using on-demand vehicles smaller than conventional buses, represents a valuable alternative for the first- and last-mile connection and in recent years, public transit agencies including the Santa Clara Valley Transportation Authority and the Alameda-Contra Costa Transit Authority (AC Transit), both in the San Francisco Bay Area, tested this model (Eno center for transportation: https://www.enotrans.org/eno -resources/uprouted-exploring-microtransit-united-states/, accessed on 14 April 2023). For example, the AC Transit Flex pilot program (AC transit Flex program: http://www.actransit.org/2016/07/18/ac-transit-launches-on-demand-flex-bus-service/, accessed on 14 April 2023), which serves the catchment area around two BART (Bay Area Rapid Transit) stations in Alameda County, provides an operating example of the flexible-route, on-demand last mile shuttle service contemplated in this study, but with human-driven, not automated, vehicles. The program improves service in low-density and low-demand areas while demonstrating cost neutrality and higher efficiency compared to the previous fixed-route service. On the other hand, using a model of existing public transit options and a hypothetical level 4 shared autonomous vehicle, Moorthy et al. (2017), explored the possibility of automated vehicles being used to solve the last mile problem in Ann Arbor MI, concluding that they significantly improved transit sustainability by promoting mode shifts to public transit [5]. It is worth noting that autonomous vehicles could serve as a paratransit and then represent a key value for people with disabilities [6].

Ride-hail systems are an important competitor to many public transport services, potentially replacing transit rides, but there are contrasting results in the literature (see [7–10]). In particular, Erhardt et al. (2022b) demonstrate that Transport Network Companies (TNCs) are responsible for a net ridership decline of about 10% in San Francisco from 2010 to 2015, offsetting net gains from other factors such as service increases and population growth [7]— moreover, Erhardt et al. (2022a) demonstrate that ride-hailing is the biggest contributor to transit ridership decline in the United States [8]. On the other hand, Hall et al. (2018) state that the TNC across U.S. metropolitan areas is a complement for the average transit agency, increasing ridership by five percent after two years [10]. However, Wang and Mu (2018) analyzed data of Uber ride-hail activity in Atlanta, highlighting that the whole ride-hail fleet cannot always guarantee sufficient accessibility in terms of wait time—in opposition to a well-supported public transport service [11]. Chee et al. (2020) state that automated bus service competes with existing last-mile services [12], while Zuo et al. (2020b) state that transit accessibility to jobs can be improved with bicycles as the first-and-last mile mode [13]; however, Rupi et al. (2019), Schweizer et al. (2020,) and Rupi et al. (2020) found that cyclists try to avoid riding in the presence of buses because otherwise this impacts cyclist safety, speed, and waiting times. Therefore, automated shuttles may offer a more convenient mode for first- and last-mile service to transit (see [14–16]).

In order to complement the current public transport service and not compete with it, automated shuttle operation should be limited to within a predefined, geographically-bounded "catchment areas", but it is not clear how to define these; a general methodology to determine the appropriate size or shape of such catchment areas is not present in the literature yet, and likely depends on the population or employment density where transit stations are located. Biba et al. (2010) used a parcel-network method for estimating the population with walking access to bus stop locations using spatial and aspatial data (i.e., location and demographics) and the network distances from parcels to bus stop locations in order to deploy the catchment area in a 100-square-mile portion of the Dallas Area Rapid Transit (DART) system covering two Texas communities [17]. Guerra et al. (2012) defined half a mile as the maximum distance from a transit stop or station that makes a

public transport service desirable to users [18]. El-Geneidy et al. (2014) declared that the 85th percentile walking distance to bus transit service is around 524 m for home-based trip origins [19]. Eom et al. (2019) found that about 90% of transit passengers traveled within 3.6 km from railway stations, suggesting a catchment area with a radius of less than 4 km [20]. AC Transit recommended service zones of approximately 5–7 square miles (AC Transit: http://www.actransit.org/flex/, accessed on 14 April 2023). In any case, as stated by Lin et al. (2019), the catchment area can increase commensurate with changes to first- and last-mile service: for example, they found that a dock-less bike sharing system can extend the catchment area beyond that when only walking is available [21]. However, it is important to take into account, as stated by Roy and Basu (2020), that poor first- and last-mile performance is generally observed in more suburban locations with low population and employment density, poor sidewalk and bus stop infrastructure, and the extra cost and time needed to make longer first- and last-mile trips using feeder buses contribute to low first- and last-mile quality [22]. It is worth noting that only the simulation of a large-scale study area allows to consider the overall impact of new elements in the transport system (see [23]). The main goal of this study is to test the operations of a fleet of shared, automated, and electric vehicles (SAEVs) integrated with existing fixed-route public transportation service in a mesoscopic simulation of a real case and large-scale scenario, while testing the Behavior, Energy, Autonomy, and Mobility beta software (BEAM) beta software capabilities.

The paper is organized as follows. Section 2 describes the methods and a case study. Section 3 shows and discusses the results of the analysis. Concluding remarks and future research directions are presented in Section 4.

## 2. Materials and Methods

The beta software BEAM—The Modeling Framework for Behavior, Energy, Autonomy, and Mobility (BEAM website: https://transportation.lbl.gov/beam, accessed on 14 April 2023)—has been under development for some years at Lawrence Berkeley National Laboratory (Lawrence Berkeley National Lab: https://www.lbl.gov/, accessed on 14 April 2023) and allows peforming large-scale and agent-based mesoscopic simulations of transport systems. The BEAM developers recently calibrated and validated a traffic scenario related to the San Francisco Bay Area. It is worth noting that the scenario has been calibrated and can be simulated only considering 10% of the population; otherwise, it would be too expensive in terms of computational power and time. Transit and road capacities have been scaled accordingly. The methodology of this study consists of developing a strategy to introduce an on-demand automated system composed of shuttles as a support for the first- and last-mile connectivity of public transportation service on BEAM, and simulate this new transportation service in Santa Clara County of the San Francisco Bay Area, which contains the city of San José and many of the communities that make up Silicon Valley (see Figure 1). In this way, it is possible to test the service and quantify the impacts of such a transportation system in a real context, as well as to test the BEAM capabilities.

### 2.1. San Francisco Bay Area Mesoscopic Model

BEAM, together with the San Francisco Bay Area scenario are available from an open source GitHub repository (BEAM GitHub repository: https://github.com/LBNL-UCB-STI/beam, accessed on 14 April 2023), and there is an online documentation (BEAM online documentation: https://beam.readthedocs.io/en/latest/, accessed on 14 April 2023). The entire Bay Area covers nine counties, is 18,000 square kilometers in area, and hosts almost eight million inhabitants, with an average density of more than 400 people per square kilometer. The scenario uses the characteristics of the road network from OpenStreetMap (OSM: https://www.openstreetmap.org/#map=5/38.007/-95.844, accessed on 14 April 2023) and adapts them to the Rapid Realistic Routing on Real-world and Reimagined networks engine (R5) requirements for multimodal routing, and contains activity-based and agent-based demand created by UrbanSim (UrbanSim: https://urbansim.com/, accessed

on 14 April 2023) for a random 10% of the population, while BEAM addresses the mode choice and the event-based mesoscopic simulation models. Each traveler (or agent) can choose via a probabilistic assignment to use a private car, either private or pooled ride-hail vehicles, public transportation service, bike, walk, or a combination of public transportation service with a walk, car, or ride-hail trip. The mode of transportation is chosen based on a within-day evaluation using a multinomial logit choice function with mode-specific value of times, and it is refined through an iterative process that consists of running the simulation several times until an equilibrium is reached, allowing for agents to revise their mode choice to better achieve their planned schedule of activities after accounting for all other agents' travel. The simulation of a general scenario with BEAM is possible via a configuration text file, where all input file paths and parameters are specified. The simulation is based on a FIFO (First In First Out) queue model on a capacity-based transport network—running the Bay Area scenario on a 10% sample for ten iterations on a high-performance computer station requires about one day of computational time. All simulation outputs are written to an output table in which each row is related to specific events that make up the legs of a single trip. The events can either be behavioral events or movement events; an example of a trip-based event chain containing the most common events is as follows: *ActivityEndEvent, ModeChoiceEvent, PersonDepartureEvent, PersonEntersVehicleEvent, PathTraversalEvent, PersonLeavesVehicleEvent, PersonArrivalEvent, ActivityStartEvent.*

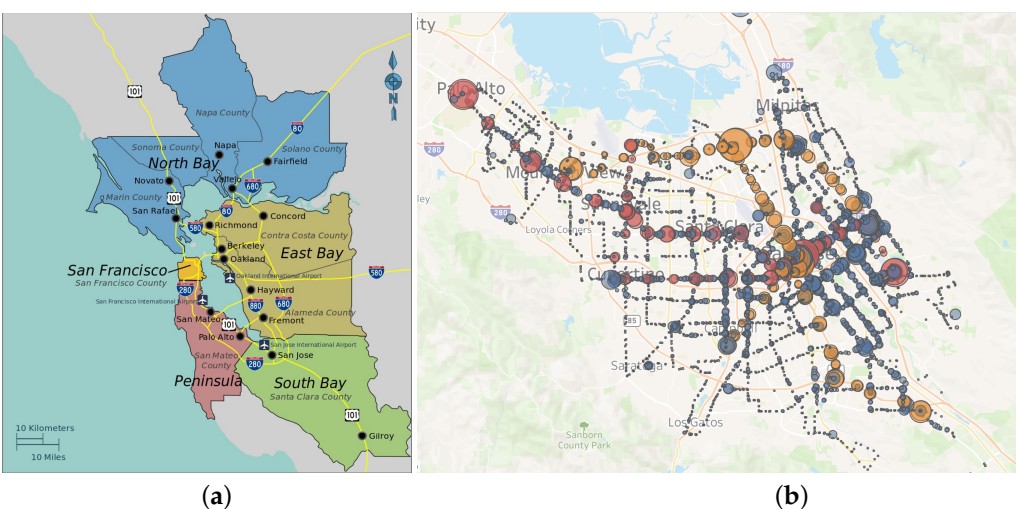

(**a**)            (**b**)

**Figure 1.** (**a**) The Bay Area and its nine counties are shown (Mapping the Bay: https://exhibits.lib.berkeley.edu/spotlight/mapthebay/feature/sf-bay-area, accessed on 14 April 2023), (**b**) The location of transit stops weighted based on the average ridership in Santa Clara County are shown—blue are bus stops, orange are light rail stations, and red are rapid bus stops (VTA transit stops ridership: https://data.vta.org/pages/ridership-by-stop, accessed on 14 April 2023).

### 2.2. Definition of the Automated Shuttles Strategy

The on-demand fleet of automated shuttles we want to simulate is characterized by passenger vans with a capacity of about 12 people, a maximum speed of 40 km/h, and in our scenario has been introduced in Santa Clara County, CA. Chee et al. (2020) declare that frequency of the primary transit routes is critical to the last-mile automated bus service usage [12]—for this reason, the strategy related to the introduction of this transportation service in our study is to provide a fleet composed of a certain number of vehicles that are constrained to operate within a circular area centered on transit stops characterized by a high frequency of arriving transit vehicles, with the objective to extend the reach of the fixed-route transit system, supporting the trip's first- and last-mile connection. The transit stops include bus stops as well as light rail and rail stations.

For the case study, based on both the literature review and on realizing a hypothetical scenario with a competitive supply, we decided to introduce 20 free-of-charge vehicles for

each of the 218 high-frequency transit stops in Santa Clara County represented in Figure 2, with a capacity of 12 passengers and a maximum speed of 40 km/h. The catchment areas were then created with a radius of 3900 m around each transit stop where at least four transit vehicles arrive per hour, on average. This distance represents how far automated vehicles can travel within 15 min travel time without congestion from the transit stops based on an isochrone analysis using ArcGIS software.

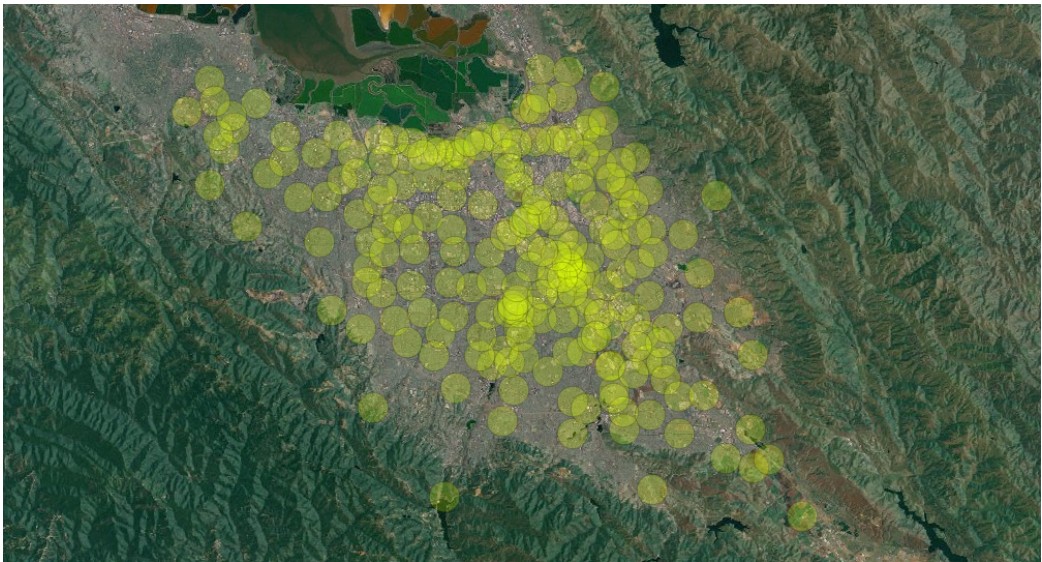

**Figure 2.** Location of high-quality transit stops considered as the center of the catchment areas for the automated shuttles in Santa Clara County, CA (Background map from Google Satellite: http://www.google.cn/maps/vt?lyrs=s@189&gl=cn&x={x}&y={y}&z={z}, accessed on 14 April 2023).

*2.3. Integrating the Automated Shuttles in the Scenario*

The strength of using the BEAM model to simulate automated shuttle service is that it is already structured in such a way to host this new transportation mode without requiring to correct the already calibrated multinomial logit mode choice model, thus gaining in both time and accuracy. In fact, the integration of this new transportation mode is mainly based on the introduction of new ride hail vehicles, but with different characteristics; that is, of larger size, of lower speed, electric, and with limited operating range (within the catchment areas). The new vehicles were manually inserted in the input files of BEAM, thus specifying both vehicle characteristics such as consumption, capacity, maximum speed, and so on, and the catchment area location and radius. The catchment area constraint was then inserted in the BEAM model directly from the software developers at Lawrence Berkeley National Laboratory (LBNL). As for the mode choice, the utility function structure of the new transportation system has been assumed the same as for the classic ride hail system, but zeroing the monetary fare. The automated shuttle capacity and speed will instead affect the user mode choice in terms of both real-time availability during the simulation at the moment of departure, and on the travel time.

## 3. Results

The fleets of automated shuttles were simulated from 12 a.m. to 8 p.m. of a typical working day, to capture the morning and afternoon peak where most of the trips are performed, and to gain in computational time. Moreover, the simulations started at midnight in order to allow people coming far from the Santa Clara County to reach the County for the morning peak hour. The same scenario was also simulated without the shuttles in order to compare the results and better analyze the efficiency and attractiveness of this new transportation system. Successively, R-software in conjunction with QGIS software allowed detailed investigation of the output table from BEAM and filtering of the results for Santa Clara County, in order to make appropriate comparisons: Figure 3 shows

where automated shuttle departures and arrivals were concentrated during mornings and afternoons.

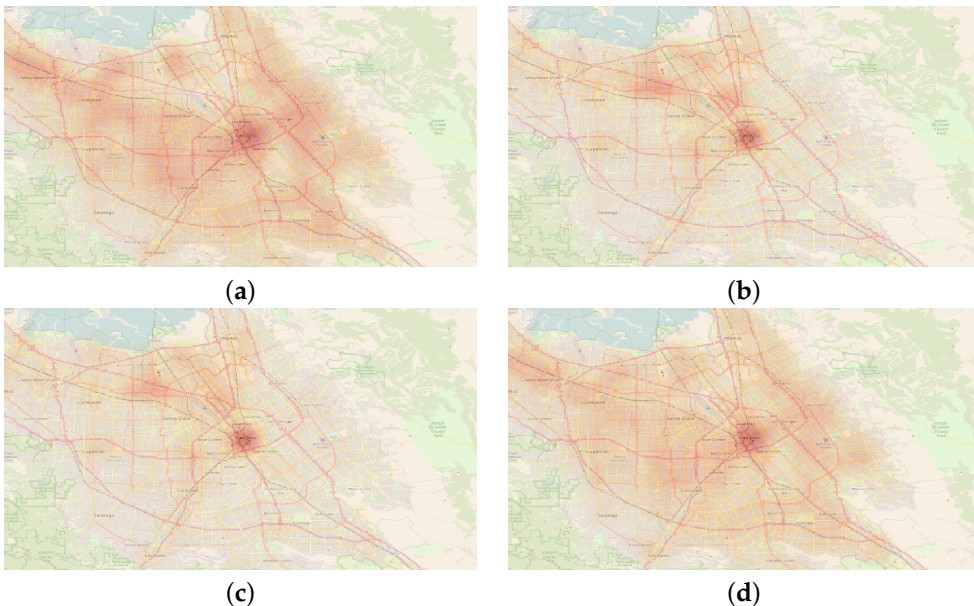

**Figure 3.** Heat Map of departures and arrivals by time of day. (**a**) Morning departures up to 12 p.m.; (**b**) Morning arrivals up to 12 p.m.; (**c**) Afternoon departures after 12 p.m.; (**d**) Afternoon arrivals after 12 p.m.

What is apparent from Figure 3 is that trips are distributed throughout the whole county, with a higher use in the downtown San José area. Morning departures (a) are more dispersed than morning arrivals (b), as many travelers presumably travel from their home to work locations in downtown San José. Afternoon arrivals (d) are slightly less dispersed than morning departures (a).

Tables 1–6, summarize the main results obtained from the post-processing analysis of the BEAM outputs filtered for the Santa Clara County, and scaled up to the full population; it is possible to observe the number of trips, length, duration, and energy consumption per each mode and fuel type, for both the baseline and the developing scenarios (see Tables 1–4), as well as a comparison of the two case studies to better understand the contribution of the new transportation mode (see Tables 5 and 6).

From Table 5, it is clear that the automated shuttles mainly replaced ride-hail trips, but also some walking trips—in particular, the ride-hail to transit trips reduced by 70%, while the overall ride-hail plus automated shuttles to transit increased by 46%, demonstrating an increasing accessibility of transit due to the automated shuttles.

The average travel length of the 11,000 legs with this new mode of transport is less than 6 km and takes an average of 9 min, due to the fact that vehicles are bounded to the catchment areas (see Table 2), while the ride hail average travel length is 25 km in the baseline (see Table 1), and goes to 27 km in the developing scenarios, where some shorter trips have been replaced by the automated vehicles (see Table 2). On the other hand, the average travel length of trips using the automated shuttles as a first- and last-mile connection is about 29 km and takes an average of 36 minutes, while it goes to 39 km and 51 min when the first- and last-mile connection is realized with a ride-hail vehicle.

It is worth noting that overall transit use increased by 3%, with a ridership increase of about 3000 trips (see Tables 1 and 2).

From Table 6, it is possible to observe how gasoline usage decreased by about 0.2% (which corresponds to about 88 million GJ), while electricity use more than tripled. While transit PMT (personal miles traveled) increased, the transit VMT (vehicle miles traveled)

remained the same, resulting in about the same amount of energy consumed in the two scenarios. On the other hand, physical energy decreased by more than 5%.

Finally, Figure 4 suggests that about 10% of automated shuttle trips occur between midnight and 5 am, probably due to the lack of other travel options during that period, which increases the accessibility of travel during night hours.

**Table 1.** Statistics of trips involving Santa Clara County in the baseline scenario per transport mode used—interfered to the population; RH = ride hail.

|  | **Trips** | **Trips [%]** | **Length [km]** | **Length [%]** | **Time [h]** | **Time [%]** | **Energy [GJ]** | **Energy [%]** |
|---|---|---|---|---|---|---|---|---|
| Bike | 18,850 | 0.99 | 243,440 | 0.47 | 13,780 | 1.95 | 2600 | 0.00 |
| Car | 1,687,240 | 88.79 | 47,759,800 | 92.02 | 575,160 | 81.31 | 85,570,830 | 81.95 |
| Drive Transit | 50,300 | 2.65 | 693,750 | 1.34 | 16,350 | 2.31 | 3,607,540 | 3.46 |
| RH | 62,410 | 3.29 | 1,565,570 | 3.02 | 18,940 | 2.68 | 2,769,050 | 2.65 |
| RH Transit | 3710 | 0.20 | 133,280 | 0.26 | 2490 | 0.35 | 356,770 | 0.34 |
| Walk | 20,350 | 1.07 | 188,350 | 0.36 | 38,630 | 5.46 | 2860 | 0.00 |
| Walk Transit | 57,320 | 3.02 | 1,319,720 | 2.54 | 42,000 | 5.94 | 12,105,410 | 11.59 |

**Table 2.** Statistics of trips involving Santa Clara County in the developing scenario per transport mode used—interfered to the population; RH = ride hail, AS = automated shuttles.

|  | **Trips** | **Trips [%]** | **Length [km]** | **Length [%]** | **Time [h]** | **Time [%]** | **Energy [GJ]** | **Energy [%]** |
|---|---|---|---|---|---|---|---|---|
| Bike | 20,660 | 1.09 | 257,410 | 0.5 | 14,580 | 2.05 | 2750 | 0.00 |
| Car | 1,688,370 | 88.79 | 47,794,680 | 92.01 | 577,870 | 81.29 | 85,642,770 | 81.87 |
| Drive Transit | 49,670 | 2.61 | 696,360 | 1.34 | 16,270 | 2.29 | 3,579,480 | 3.42 |
| RH | 53,710 | 2.82 | 1,461,180 | 2.81 | 18,540 | 2.61 | 2,579,180 | 2.46 |
| AS | 5640 | 0.30 | 33,200 | 0.07 | 860 | 0.12 | 44,570 | 0.05 |
| RH Transit | 1080 | 0.06 | 42,540 | 0.08 | 910 | 0.13 | 137,940 | 0.13 |
| AS Transit | 4350 | 0.23 | 124,850 | 0.24 | 2610 | 0.37 | 279,480 | 0.27 |
| Walk | 18,670 | 0.98 | 174,260 | 0.34 | 35,690 | 5.02 | 2430 | 0.00 |
| Walk Transit | 59,470 | 3.13 | 1,360,610 | 2.62 | 43,550 | 6.13 | 12,336,610 | 11.79 |

**Table 3.** Statistics of trips involving Santa Clara County in the baseline scenario per fuel type—interfered to the population

| **Fuel Type** | **Trips** | **Trips [%]** | **Length [km]** | **Length [%]** | **Duration [h]** | **Duration [%]** | **Energy [GJ]** | **Energy [%]** |
|---|---|---|---|---|---|---|---|---|
| Biodiesel | 89,610 | 4.24 | 704,400 | 1.36 | 23,990 | 3.43 | 12,709,770 | 12.30 |
| Diesel | 50,350 | 2.38 | 1,171,090 | 2.27 | 19,540 | 2.79 | 2,311,570 | 2.24 |
| Electricity | 21,910 | 1.04 | 220,760 | 0.43 | 5990 | 0.86 | 38,780 | 0.04 |
| Food | 159,790 | 7.56 | 256,010 | 0.50 | 54,880 | 7.85 | 6320 | 0.01 |
| Gasoline | 1,792,440 | 84.79 | 49,286,680 | 95.44 | 594,740 | 85.07 | 88,243,340 | 85.42 |

**Table 4.** Statistics of trips involving Santa Clara County in the developing scenario by fuel type—interfered to the population.

| **Fuel Type** | **Trips** | **Trips [%]** | **Length [km]** | **Length [%]** | **Duration [h]** | **Duration [%]** | **Energy [GJ]** | **Energy [%]** |
|---|---|---|---|---|---|---|---|---|
| Biodiesel | 91,420 | 4.31 | 711,850 | 1.38 | 24,460 | 3.48 | 12,844,190 | 12.41 |
| Diesel | 51,550 | 2.43 | 1,213,580 | 2.35 | 20,330 | 2.89 | 2,409,290 | 2.33 |
| Electricity | 33,900 | 1.60 | 293,970 | 0.57 | 7950 | 1.13 | 125,370 | 0.12 |
| Food | 160,080 | 7.55 | 245,750 | 0.48 | 52,890 | 7.53 | 5970 | 0.01 |
| Gasoline | 1,782,550 | 84.10 | 49,203,780 | 95.23 | 596,620 | 84.96 | 88,087,040 | 85.13 |

**Table 5.** Statistics of trips involving Santa Clara County: differences between the baseline and the developing scenarios per transport mode used—interfered to the population; RH = ride hail.

|  | Trips [Δ] | Trips [Δ%] | Length [Δ Km] | Length [Δ Km %] | Time [Δ h] | Time [Δ h %] | Energy [Δ GJ] | Energy [Δ GJ %] |
|---|---|---|---|---|---|---|---|---|
| Bike | 1810 | 9.60% | 13,970 | 5.74% | 800 | 5.81% | 150 | 5.77% |
| Car | 1130 | 0.07% | 34,880 | 0.07% | 2710 | 0.47% | 71,940 | 0.08% |
| Drive Transit | −630 | −1.25% | 2610 | 0.38% | −80 | −0.49% | −28,060 | −0.78% |
| RH | −8700 | −13.94% | −104,390 | −6.67% | −400 | −2.11% | −189,870 | −6.86% |
| RH Transit | −2630 | −70.89% | −90,740 | −68.08% | −1580 | −63.45% | −218,830 | −61.34% |
| Walk | −1680 | −8.26% | −14,090 | −7.48% | −2940 | −7.61% | −430 | −15.03% |
| Walk Transit | 2150 | 3.75% | 40,890 | 3.10% | 1550 | 3.69% | 231,200 | 1.91% |

**Table 6.** Statistics of trips involving Santa Clara County: differences between the baseline and the developing scenario per fuel type—interfered to the population.

| Fuel Type | Trips [Δ] | Trips [Δ%] | Length [Δ Km] | Length [Δ Km %] | Duration [Δ h] | Duration [Δ h %] | Energy [Δ GJ] | Energy [Δ GJ %] |
|---|---|---|---|---|---|---|---|---|
| Biodiesel | 1810 | 2.02% | 7450 | 1.06% | 470 | 1.96% | 134,420 | 1.06% |
| Diesel | 1200 | 2.38% | 42,490 | 3.63% | 790 | 4.04% | 97,720 | 4.23% |
| Electricity | 11,990 | 54.72% | 73,210 | 33.16% | 1960 | 32.72% | 86,590 | 223.29% |
| Food | 290 | 0.18% | −10,260 | −4.01% | −1990 | −3.63% | −350 | −5.54% |
| Gasoline | −9890 | −0.55% | −82,900 | −0.17% | 1880 | 0.32% | −156,300 | −0.18% |

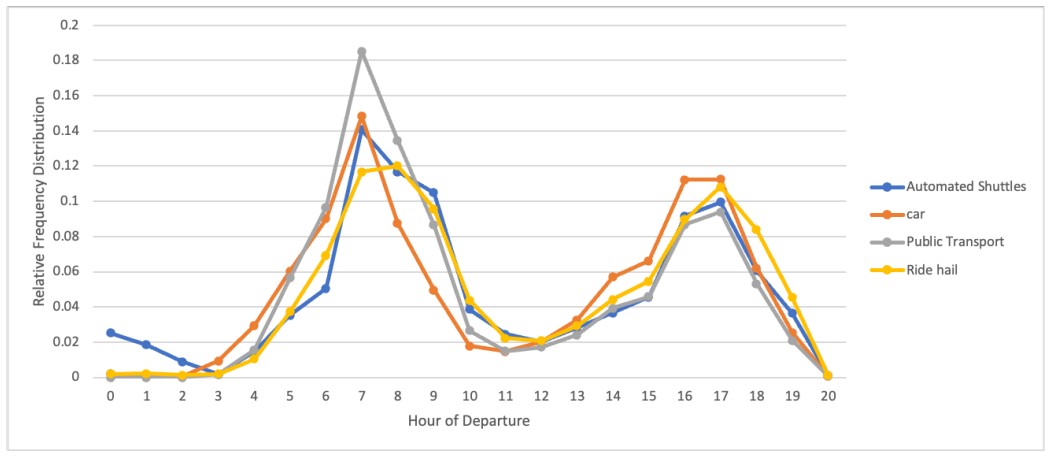

**Figure 4.** Relative frequency distribution of departure time of automated shuttles during the day, compared with the usage of cars, other ride-hails, and public transport service.

## 4. Conclusions

The main goal of the present study was to evaluate the impacts of a fleet of automated shuttles in Santa Clara County providing first- and last-mile connections to the current public transportation system. The secondary goal was to use and test the LBNL BEAM beta software capabilities. BEAM was selected for use in this study because it demonstrated the capability to simulate the proposed automated shuttle service as a combination of several critical features: defining a new vehicle type approximating the automated electric shuttle characteristic and performance; assigning a ride hail manager and utility function to the new shuttles; establishing geofenced areas throughout a broad area such as a county, and assigning these to the shuttles. Regarding the application to the Santa Clara County scenario, the model suggests that automated shuttles would be chosen by some users for either first- and last-mile connections to transit or as a single mode to reach a nearby destination. This new mode has proven to mostly replace short-distance ride-hail connections, especially those directed to transit stops, but also

walking trips. The presence of automated shuttles in the scenario has brought several benefits: decrease on gasoline-base trips, overall increase in transit usage, and first- and last-mile connections to transit, proving a higher accessibility of transit, especially during night hours. It is also true that an electric vehicle fleet requires particular attention to the interaction with the electric grid: e.g. charging time and charging spot availability. In fact, it is important to highlight that a limitation of this study is represented by not considering this layer of detail as the implementation of the electric grid modelling is still under calibration on BEAM: therefore, this study might either overestimate the ridership attracted by the automated shuttles, or underestimate the number of vehicles needed to achieve the observed ridership, as they are assumed to be running non-stop. For these reasons, from an operational and economic point of view, it is necessary to take into consideration an eventual oversized fleet and quick recharging to maximize the vehicles' operational time in a day (see [24–26]), as well as the implication for the potential revenue shift from privately-run TNCs to public agencies. It is worth noting that the share of car trips in the area is about 89% (see Table 1), which means it is a car-based area where it will be very difficult to attract people to other environmentally safe modes; however, even if results are eclipsed by the massive presence of cars, it is still possible to observe some overall improvements due to the presence of automated shuttles in the Bay Area. Future research could provide further insight into the potential use of automated shuttles for connectivity, as well as integrate the electric grid effect in the simulation. The main benefit of this work has been demonstrating that the BEAM model, despite its beta status, allowed for the analysis of a large-scale scenario in quite a high level of detail. As the BEAM model continues to be developed, it will become more robust and capable in the future. The BEAM developers have been very open to inserting new tools, parameters, and functions to adapt BEAM to this case study. Moreover, the BEAM model is very flexible for analyzing different interventions on the baseline scenario and it could clearly be adapted to a multitude of different studies.

**Author Contributions:** Conceptualization, Cristian Poliziani, Gary Hsueh, David Czerwinski and Joerg Schweizer; methodology, Cristian Poliziani, Gary Hsueh and David Czerwinski; software, Cristian Poliziani, Zachary Needell and Haitam Laarabi; data curation, Cristian Poliziani, Gary Hsueh and David Czerwinski; writing—original draft preparation, Cristian Poliziani, Gary Hsueh and David Czerwinski; writing—review and editing, Cristian Poliziani, Gary Hsueh, David Czerwinski, Tom Wenzel, Zachary Needell, Haitam Laarabi, Joerg Schweizer and Federico Rupi; visualization, Cristian Poliziani; supervision, Cristian Poliziani, Gary Hsueh, David Czerwinski, Tom Wenzel, Joerg Schweizer and Federico Rupi; project administration, Gary Hsueh and David Czerwinski; funding acquisition, Gary Hsueh and David Czerwinski All authors have read and agreed to the published version of the manuscript.

**Funding:** This research was funded by United States Department of Transportation (DOT) grant number 69A3551747127.

**Data Availability Statement:** The data supporting reported results can be found at https://scholarworks.sjsu.edu/cgi/viewcontent.cgi?filename=1&article=1341&context=mti_publications&type=additional accessed on 14 April 2023.

**Acknowledgments:** The authors would like to thank Richard Kos of San José State University and several students from his Fall 2018 Advanced GIS Class—Frank Arellano, Matthew Moore, and Michael Mulligan—for their background research and GIS analysis. The authors also thank Colin Sheppard and BEAM developers at Lawrence Berkeley National Laboratory for their support with BEAM. The authors thank MTI staff, including Karen Philbrick, Hilary Nixon, editing staff, and peer reviewers for thoughtful comments and support during the progress of the MTI research project [27] from which this article has been developed.

**Conflicts of Interest:** The authors declare no conflict of interest. The funders had no role in the design of the study; in the collection, analyses, or interpretation of data; in the writing of the manuscript; or in the decision to publish the results.

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
