# Peer review of "Micro Transit Simulation of On-Demand Shuttles Based on Transit Data for First- and Last-Mile Connection"

_ijgi, doi:10.3390/ijgi12040177_

Round 1

Reviewer 1 Report

The main goal of the research was to evaluate the impacts of a fleet of automated shuttles, providing first and last mile connections to the current public transportation system. The topic is relevant,  the used methods are consistent with the aim of the research and the results are well described.

Author Response

Authors thanks the Reviewer 1 for their great words and for the review.

Reviewer 2 Report

      Page 2, line 42, “(AC Transit)” should be added after “Alameda-Contra Costa Transit Authority” as it is shown in later context.

2.     Figure 1(b), orange and red are too similar to tell the difference between two types of nodes. There is no explanation about what “rapid” is.

3.     Page4, line 122, why only 10% population is applied? After simulation, does the results be scaled to the whole population, or the results only present 10% (maybe lower than 10%) of the outcomes, like ridership and traffic?

4.     Page 5, line 179,  “The fleets of automated shuttles were simulated from 12 AM to 8 PM of a typical 179 working day”. There are some late trips from work to home and some social/recreation trips going back home lately during 8pm-midnight. Under this setting, such trips are not simulated.

5.     In Section 2.3, the utility function of automated shuttle is the same as ride hail except with zero fare. The authors should have more explanation why using this assumption. Automated shuttle should have longer travel time as it is 12-passenger shuttle and is not from transit stop directly to home unless there is no other passenger in shuttle.

6.     It makes sense that most ride hail trips go to automated shuttle due to AS’s zero fare. But this means the cost of first and last miles is shifted to public agencies. This is another topic, but the authors should address this assumption and have some discussions in conclusion.

7.     Overall, the authors address and simulate the one-demand shuttle for first mile and last mile connection successfully but they need to address several assumptions and questions proposed above.

Author Response

Page 2, line 42, “(AC Transit)” should be added after “Alameda-Contra Costa Transit Authority” as it is shown in later context.

Authors thanks for the suggestion, the text has been changed accordingly.

  1. Figure 1(b), orange and red are too similar to tell the difference between two types of nodes. There is no explanation about what “rapid” is.

Authors agree with the Reviewer. Unfortunately, the map comes from the provided link in the footnote, and it’s made by the Valley Transportation Authority (VTA): we have not access to it. However, the meaning of rapid has been better explained: it means served by the so-called rapid buses from VTA, and they define as “limited-stop service at frequent intervals”

  1. Page4, line 122, why only 10% population is applied? After simulation, does the results be scaled to the whole population, or the results only present 10% (maybe lower than 10%) of the outcomes, like ridership and traffic?

The scenario has been calibrated and can be simulated only considering the 10% of the population, otherwise it would be too expensive in terms of computational power and time:  transit and road capacities have been scaled accordingly. The results have been scaled up to the full population – this has been better explained along the text:

It is worth noting that the scenario has been calibrated and can be simulated only considering the 10\% of the population, otherwise it would be too expensive in terms of computational power and time: transit and road capacities have been scaled accordingly.

Tables 1, 2, 3, 4, 5 and 6 summarize the main results obtained from the post-processing analysis of the BEAM outputs filtered for the Santa Clara County, and scaled up to the full population

  1. Page 5, line 179,  “The fleets of automated shuttles were simulated from 12 AM to 8 PM of a typical 179 working day”. There are some late trips from work to home and some social/recreation trips going back home lately during 8pm-midnight. Under this setting, such trips are not simulated.

Reviewer is right. The run time has been cut off at 8pm since Authors were mainly interested in the morning and afternoon peak (see figure 4), where most of the trips are performed, and to gain in computational time. Moreover, the simulations started at midnight in order to warm up the transport network with people coming far from the Santa Clara County, allowing them to reach the County for the morning peak hour. This has been better explained along the text:

‘’

The fleets of automated shuttles were simulated from 12 AM to 8 PM of a typical working day: doing that it has been possible to capture the morning and afternoon peak where most of the trips are performed, and to gain in computational time. Moreover, the simulations started at midnight in order to warm up the transport network with people coming far from the Santa Clara County, allowing them to reach the County for the morning peak hour.

‘’

  1. In Section 2.3, the utility function of automated shuttle is the same as ride hail except with zero fare. The authors should have more explanation why using this assumption. Automated shuttle should have longer travel time as it is 12-passenger shuttle and is not from transit stop directly to home unless there is no other passenger in shuttle.

As for the mode choice, the utility function structure of the new transportation system has been assumed the same as for the classic ride hail system, but zeroing the monetary fare. This has been better explained along the text:

The catchment area constraint has been then inserted in the BEAM model directly from the software developers at LBNL. As for the mode choice, the utility function structure of the new transportation system has been assumed the same as for the classic ride hail system, but zeroing the monetary fare. The automated shuttle capacity and speed will instead affect the user mode choice in terms of both real-time availability during the simulation at the moment of departure, and on the travel time.

Travel durations appear shorter for the automated shuttles (≈9’) than for the classical ride hail (≈20’), as expected, mostly because the automated shuttles are bounded to serve within a small catchment area. Therefore, even if they might have several passengers to pick up and drop off along the path, the total duration for the automated shuttle users will be lower than for classical ride hail, for which there is not such a restricted boundary.

  1. It makes sense that most ride hail trips go to automated shuttle due to AS’s zero fare. But this means the cost of first and last miles is shifted to public agencies. This is another topic, but the authors should address this assumption and have some discussions in conclusion.

Reviewer is right. Authors thanks for raising this point, and have added an appropriate discussion in the conclusions:

The presence of the automated shuttles in the scenario has brought several benefits: decrease on gasoline-base trips, overall increase of transit usage and first and last mile connections to transit, proving a higher accessibility to transit, especially during night hours.

It is also true that an electric vehicle fleet requires particular attention to the interaction with the electric grid: e.g. charging time, charging spot availability. In fact, it is important to highlight that a limitation of this study is represented by not considering this layer of detail as the implementation of the electric grid modelling is still under calibration on BEAM: therefore, this study might either overestimate the ridership attracted by the automated shuttles, or underestimate the number of vehicles needed to achieve the observed ridership, as they are assumed to be running non-stop. For these reasons, from an operational and economic point of view it is necessary to take into consideration an eventual oversized fleet and quick recharging to maximize the vehicles' operational time in a day (see \cite{Huang} and \cite{Gerboni}), as well as the implication for the potential revenue shift from privately run TNCs to public agencies.

  1. Overall, the authors address and simulate the one-demand shuttle for first mile and last mile connection successfully but they need to address several assumptions and questions proposed above.

Authors thanks the Reviewer 2 for their great words and for the review that definitely improved the paper quality.

Reviewer 3 Report

The study is well set up and conducted with an adequate state of the art.

However, there seems to be an important missing piece in this type of service and therefore of related analysis: the guarantee of the service itself in relation to vehicle recharging times and availability of the recharging network, both in terms of free places and network power. The latter, for the size of the fleet, should not create a problem but certainly would in the case of scalability of the service, while actual availability of recharging stations and recharging times have two implications: the cost of electricity, in the case of fast or quick recharging, far higher than a refuelling (at current prices), and the availability of vehicles, which typically implies an oversized fleet. Thus, it is true that fuel is saved, as written in the paper, but it is equally true that the fleet requires more vehicles than those equipped with internal combustion engines. A note on these aspects should be written for completeness and current economic interest true to this service; examples from literature, not binding, just to let you better understand this comment:

https://doi.org/10.1016/j.trd.2019.11.008

https://doi.org/10.1016/j.trd.2018.02.009

https://doi.org/10.1016/j.trip.2021.100454

The first time you mention TNC declare the extended meaning.

Conclusions should include the limits of this analysis, as suggested before.  

Author Response

The study is well set up and conducted with an adequate state of the art.

However, there seems to be an important missing piece in this type of service and therefore of related analysis: the guarantee of the service itself in relation to vehicle recharging times and availability of the recharging network, both in terms of free places and network power. The latter, for the size of the fleet, should not create a problem but certainly would in the case of scalability of the service, while actual availability of recharging stations and recharging times have two implications: the cost of electricity, in the case of fast or quick recharging, far higher than a refuelling (at current prices), and the availability of vehicles, which typically implies an oversized fleet. Thus, it is true that fuel is saved, as written in the paper, but it is equally true that the fleet requires more vehicles than those equipped with internal combustion engines. A note on these aspects should be written for completeness and current economic interest true to this service; examples from literature, not binding, just to let you better understand this comment:

https://doi.org/10.1016/j.trd.2019.11.008

https://doi.org/10.1016/j.trd.2018.02.009

https://doi.org/10.1016/j.trip.2021.100454

The first time you mention TNC declare the extended meaning.

Conclusions should include the limits of this analysis, as suggested before.  

Authors thanks the Reviewer for their great words and for the review that definitely improved the paper quality. Unfortunately, the electric grid modelling is not ready to be incorporated on the BEAM model yet, but as the Reviewer correctly suggested, it would add another level of detail to this study. The BEAM developers are currently working on that and future research might want to add this layer to this study.

The following the has been added:

The presence of the automated shuttles in the scenario has brought several benefits: decrease on gasoline-base trips, overall increase of transit usage and first and last mile connections to transit, proving a higher accessibility to transit, especially during night hours.

It is also true that an electric vehicle fleet requires particular attention to the interaction with the electric grid: e.g. charging time, charging spot availability. In fact, it is important to highlight that a limitation of this study is represented by not considering this layer of detail as the implementation of the electric grid modelling is still under calibration on BEAM: therefore, this study might either overestimate the ridership attracted by the automated shuttles, or underestimate the number of vehicles needed to achieve the observed ridership, as they are assumed to be running non-stop. For these reasons, from an operational and economic point of view it is necessary to take into consideration an eventual oversized fleet and quick recharging to maximize the vehicles' operational time in a day (see [23] and [24]) as well as the implication for the potential revenue shift from privately run TNCs to public agencies.

The authors also thank the reviewer 3 for the suggested articles that were deemed worthy of citation in our article to enrich the literature review. Finally, the TNC acronym has been correctly explained in the text.
